# Efficient and practical Hamiltonian simulation from time-dependent product formulas

Jan Lukas Bosse[1,2], Andrew M. Childs[1,3], Charles Derby[1], Filippo Maria Gambetta ®[1], Ashley Montanaro[1,2] & Raul A. Santos ®[1] ✉

In this work we propose an approach for implementing time-evolution of a quantum system using product formulas. The quantum algorithms we develop have provably better scaling (in terms of gate complexity and circuit depth) than a naive application of well-known Trotter formulas, for systems where the evolution is determined by a Hamiltonian with different energy scales (i.e., one part is "large" and another part is "small"). Our algorithms generate a decomposition of the evolution operator into a product of simple unitaries that are directly implementable on a quantum computer. Although the theoretical scaling is suboptimal compared with state-of-the-art algorithms (e.g., quantum signal processing), the performance of the algorithms we propose is highly competitive in practice. We illustrate this via extensive numerical simulations for several models. For instance, in the strong-field regime of the 1D transverse-field Ising model, our algorithms achieve an improvement of one order of magnitude in both the system size and evolution time that can be simulated with a fixed budget of 1000 arbitrary 2-qubit gates, compared with standard Trotter formulas.

Time-dynamics simulation (TDS) of quantum systems has long been considered as a natural application where quantum computers can outperform classical ones. A quantum algorithm for TDS approximates the time-evolution operator $e^{-itH}$ by a sequence of elementary gates. The gate complexity of this decomposition is at least linear in $t$ in general[1,2], and several methods have been proposed that achieve (or nearly achieve) that complexity[3–6]. These methods differ in the way they implement time evolution, have different overheads, and scale differently with the desired accuracy.

Arguably the most straightforward TDS algorithm is the use of (Trotter) product formulas[7]. This approach does not use ancilla qubits[3–5], nor does it involve potentially costly operations (such as block encodings or reflections about ancillary quantum states)[6], or any classical pre-processing (such as searching for classically optimized circuits[8,9]). Moreover, product formulas can be more efficient in

practice when simulating systems with hundreds of qubits for times that scale with the size of the system[10]. This may be due to overheads that some asymptotically better algorithms incur, and to the fact that product formula methods scale better in practice than naive bounds suggest, with dependence on commutators of terms that can naturally take advantage of spatial locality[11,12].

Product formulas split the evolution under a Hamiltonian $H = \sum_k h_k$ into a product of the form $\prod_{jk} e^{-it_{jk}h_k}$ for some times $t_{jk}$. This provides an efficient simulation if each elementary exponential can be implemented efficiently. Observe that the choice of the summands that compose $H$ is not unique. A common practice when simulating lattice systems is to represent the Hamiltonian as a sum of Pauli terms $H = \sum_k \alpha_k P_k$ and choose $h_k = \alpha_k P_k$.

Starting from a Hamiltonian generating time evolution that can be implemented with a quantum circuit with error independent of the

[1]Phasecraft Ltd. 77 Charlotte Street, W1T 4PW London, UK. [2]School of Mathematics, University of Bristol, Bristol, UK. [3]Department of Computer Science, Institute for Advanced Computer Studies, Joint Center for Quantum Information and Computer Science, University of Maryland, College Park, MD, USA. ✉e-mail: raul@phasecraft.io

evolution time (e.g., a Hamiltonian diagonal in the computational basis, or diagonalisable with a circuit that does not scale with the evolution time), we ask, "What is the effect of adding a perturbation to the Hamiltonian on the complexity of implementing the TDS algorithm with product formulas?". This motivates going into the interaction picture and approximating the time-ordered operator by a product of exponentials.

This technique allows us to introduce several algorithms that take advantage of the structure of the Hamiltonian to achieve better error scaling than standard product formulas. This approach can leverage knowledge of the gates that can be efficiently implemented in practice on a particular quantum computer, so we call this family of algorithms Trotter Heuristic Resource Improved Formulas for Time-dynamics (THRIFT). The interaction-picture approach has been studied in ref. [13], where approximations of the time-ordered operator are done through a Taylor expansion of the Dyson series, instead of using product formulas achieving a gate complexity of $O(\alpha T \operatorname{polylog}(T\alpha/\epsilon))$ for a simulation for time $T$ with error $\epsilon$. Although, theoretically, the LCU method has better scaling with evolution time and simulation error than product formulas, it has also been shown empirically that product-formula approaches can perform better in practice[10]. Furthermore, the LCU method uses ancilla qubits and involves implementing both an operation that coherently performs the constituent unitaries conditioned on the ancilla and a reflection about a certain ancilla state. Our approach uses no ancillas and only involves evolution according to terms of the Hamiltonian, as it directly implements the time evolution using product formulas, achieving a gate complexity of $O(\alpha T(\alpha T/\epsilon)^{1/(k-1)})$ for arbitrary fixed $k$.

Using Lieb–Robinson bounds ref. [14] introduces a protocol for quantum simulation of lattice models that resembles the THRIFT algorithm described in Eq. (7), but where the splitting of the Hamiltonian is decided based on the support of its summands, not on the energy scales involved in the Hamiltonian. The cost of this method is nearly optimal as a function of system size as well as evolution time and approximation error. However, in practice, this strategy may perform worse than straightforward application of product formulas[11].

After completing an initial version of this work, we became aware of ref. [15]. There, Omelyan et al. derive a set of optimised fourth-order product formulas for a Hamiltonian $H = H_A + H_B$ by adding additional sub-steps to standard Trotter formulas and numerically minimizing the error arising from commutators (see Supplementary Note 3 for details). The generalisation to Hamiltonians with an arbitrary number of terms is described in ref. [7]. Of particular interest for the present work is an optimised formula valid in the regime $\alpha \ll 1$ of a Hamiltonian $H = H_0 + \alpha H_1$ (denominated "Omelyan's small A" in ref. [7]). In Results we show that for the 1D and 2D transverse-field Ising and the 1D Heisenberg models, THRIFT outperforms this optimised formula for all the values of $\alpha$ we consider. On the other hand, Omelyan et al.'s optimised small A formula proves to be the most efficient algorithm in the $t_{\text{hop}}/U \ll 1$ regime of the Fermi–Hubbard model. This is mainly due to the high cost of implementing the terms arising in the THRIFT decomposition of this model.

In this work, we generate an efficient product-formula decomposition for time evolution of a quantum system. This decomposition has provably better scaling of both gate complexity and circuit depth than a naive application of well-known product formulas, for systems where the evolution is determined by a Hamiltonian with different energy scales (i.e., in which one part is "large" and another part is "small", with the size of the small part quantified by a parameter $\alpha$). This situation is ubiquitous in effective models describing physical systems and can occur, for example, in systems with strong short-range interactions and weaker long-range interactions. Furthermore, weak (or strong) external perturbations can be used to push a system out of equilibrium and to extract its dynamical properties. Crucially,

the efficiency of the algorithm depends on the characteristics of the quantum computer itself, namely, the set of gates that are easily implementable with an error independent of the circuit depth. This is particularly useful in a noisy intermediate-scale quantum (NISQ) computer, where some types of gates can be implemented more easily than other nominally similar gates. As these formulas provide better gate complexity than naive product formulas in many instances, we expect them to be useful beyond NISQ applications as well.

In "Results", we introduce THRIFT and show that its error scales as $O(\alpha^2 t^2)$, an improvement by a factor of $\alpha$ compared with standard first-order product formulas. We show that $k$th-order THRIFT achieves error-scaling of $O(\alpha^2 t^{k+1})$, compared to $O(\alpha t^{k+1})$ for standard $k$th-order formulas. We also show (in Supplementary Note 1) that general product formulas based directly on products of the summands of the Hamiltonian cannot achieve better scaling than $\alpha^2$. To improve the $\alpha$-scaling for higher-order formulas, we introduce the Magnus-THRIFT and Fer-THRIFT algorithms in "Results" and Supplementary Note 2, respectively, which achieve an effective $O(\alpha^{k+1} t^{k+1})$ error scaling, for any $k \in \mathbb{N}$. These results are valid for partitions of the Hamiltonian where implementing a piece of the perturbation together with the dominant Hamiltonian does not incur in an error larger than $O(\alpha^2)$, as we discuss below.

To complement our theoretical results that show favourable asymptotic scaling of the algorithms, in Results we carry out numerical experiments comparing several product formulas with THRIFT. We analyse the error as a function of the total evolution time and the scale of the small part of the Hamiltonian $\alpha$ for three different models: the transverse-field Ising model in one (1D) and two dimensions (2D), the 1D Heisenberg model with random fields, and the 1D Fermi–Hubbard model. For the spin models studied, the THRIFT approach generates better product formulas in terms of gate complexity (measured as the number of CNOT or arbitrary 2-qubit gates to achieve a target error) for a wide range of evolution times and $\alpha$. We stress that, despite THRIFT formulas having been derived assuming a small $\alpha$, our numerical results show that in the case of the transverse-field Ising and Heisenberg models, THRIFT formulas outperform standard product formulas also for intermediate and large values of $\alpha$. Despite the simplicity of such models, they correctly describe the relevant physics of a wide range of low-dimensional magnetic materials and, in the presence of frustration, can host exotic quantum phases of matter[16–18]. In this context, the introduction of perturbative corrections is often required in order to obtain a precise agreement with experimental data. A paradigmatic example is the quasi-1D ferromagnet $CoNb_2O_6$, which is regarded as one of the closest experimental realizations of the 1D transverse-field Ising model. However, to describe the complex physics emerging near its critical point, one has to consider a more detailed microscopic model containing several perturbations, which break the integrability of the Hamiltonian and make numerical simulations more challenging[19–21]. In the simulation of the transverse-field Ising and Heisenberg models, the favourable scaling is due to the possibility of implementing the elementary evolution gates with a 2-qubit gate cost that is the same as standard product formulas. For simulations of the Fermi-Hubbard model, THRIFT methods have advantageous scaling for large enough simulation time $T \gtrsim U^{-1}$ and small scale of the hopping term $t_{\text{hop}}/U$. This is due to the extra cost incurred in the implementation of THRIFT in this case.

## Results
### THRIFT algorithms
Consider a Hamiltonian of the form $H = H_0 + \alpha H_1$ where $\alpha \ll 1$, the norms of $H_0$ and $H_1$ are comparable, and the unitary $U_0 = e^{-itH_0}$ can be implemented exactly for arbitrary times $t$ with an efficient quantum circuit, with complexity independent of $t$. We are interested in approximating the full evolution operator $U = e^{-itH}$. The first-order

Trotter formula with $N$ steps has error[7,11]

$$\| e^{-it(H_0 + \alpha H_1)} - (e^{-i\frac{t}{N}H_0} e^{-i\frac{t}{N}\alpha H_1})^N \| \leq \frac{t^2 |\alpha|}{2N} \| [H_0, H_1] \| . \quad (1)$$

We can use the fact that $U_0$ is implementable exactly to give a simulation with lower error. Going to the interaction (also known as intermediate) picture[22], we have

$$
\begin{aligned}
U &= \lim_{N \to \infty} \prod_{k=1}^{N} e^{-i\frac{t}{N}H_0} e^{-i\frac{t}{N}\alpha H_1}, \\
&= e^{-itH_0} \lim_{N \to \infty} e^{\frac{i(N-1)t}{N}H_0} e^{-i\frac{t}{N}\alpha H_1} e^{-\frac{i(N-1)t}{N}H_0} \cdots e^{-i\frac{t}{N}\alpha H_1} e^{i\frac{t}{N}H_0} e^{-i\frac{t}{N}\alpha H_1} e^{-i\frac{t}{N}H_0} e^{-i\frac{t}{N}\alpha H_1}, \\
&= e^{-itH_0} \mathcal{T} e^{-i \int_0^t \alpha \tilde{H}_1(\tau) d\tau},
\end{aligned}
\quad (2)
$$

where in the second line we have just inserted identities between each exponential of $H_1$. Here, $\mathcal{T}$ is the time-ordering operator (which moves terms with smaller times to the right) and $\tilde{H}_1(t) = e^{itH_0} H_1 e^{-itH_0}$. This is a better starting expression for bounding the error. Let $[\mathcal{T} e^{-i \int_0^t \alpha \tilde{H}_1(\tau) d\tau}]_{\text{apx}}$ denote a product formula (to be defined) for approximating $\mathcal{T} e^{-i \int_0^t \alpha \tilde{H}_1(\tau) d\tau}$, and let $U_{\text{apx}}$ denote the overall approximation to $U$ obtained by using this formula. Then we have

$$
\begin{aligned}
\| U - U_{\text{apx}} \| &= \| e^{-itH_0} \mathcal{T} e^{-i \int_0^t \alpha \tilde{H}_1(\tau) d\tau} - e^{-itH_0} [\mathcal{T} e^{-i \int_0^t \alpha \tilde{H}_1(\tau) d\tau}]_{\text{apx}} \| \\
&= \| \mathcal{T} e^{-i \int_0^t \alpha \tilde{H}_1(\tau) d\tau} - [\mathcal{T} e^{-i \int_0^t \alpha \tilde{H}_1(\tau) d\tau}]_{\text{apx}} \|
\end{aligned}
\quad (3)
$$

by invariance of the operator norm under unitary transformations. Using for example the first-order generalised Trotter formula $[\mathcal{T} e^{-i \int_0^t \alpha \tilde{H}_1(\tau) d\tau}]_{\text{apx}} = \mathcal{T} e^{-i \int_0^t \alpha \tilde{H}_1^A(\tau) d\tau} \mathcal{T} e^{-i \int_0^t \alpha \tilde{H}_1^B(\tau) d\tau}$[23,24], where $\tilde{H}_1(\tau) = \tilde{H}_1^A(\tau) + \tilde{H}_1^B(\tau)$ is some splitting of $\tilde{H}_1(\tau)$, we have

$$
\begin{aligned}
\| U - U_{\text{apx}} \| &= \| \mathcal{T} e^{-i \int_0^t \alpha \tilde{H}_1(\tau) d\tau} - \mathcal{T} e^{-i \int_0^t \alpha \tilde{H}_1^A(\tau) d\tau} \mathcal{T} e^{-i \int_0^t \alpha \tilde{H}_1^B(\tau) d\tau} \|, \\
&\leq \alpha^2 \int_0^t dv \int_0^v ds \| [\tilde{H}_1^A(s), \tilde{H}_1^B(v)] \| = O(\alpha^2 t^2) \quad \text{using}^{[23]},
\end{aligned}
\quad (4)
$$

assuming that $\| [\tilde{H}_1^A(s), \tilde{H}_1^B(v)] \| = O(1)$. Note that the error now scales as $\alpha^2$ instead of $\alpha$. The inequality in Eq. (4) can be shown by writing the integral representation of the error for the time-dependent case. This forms the basis for the proof presented in[23]. For general evolution time, we can divide the evolution into $N$ steps, giving an error

$$
\begin{aligned}
\| U - U_{\text{apx}} \| &= \| \mathcal{T} e^{-i \int_0^t \alpha \tilde{H}_1(\tau) d\tau} - \prod_{j=0}^{N-1} \mathcal{T} e^{-i \int_{j\frac{t}{N}}^{(j+1)\frac{t}{N}} \alpha \tilde{H}_1^A(\tau) d\tau} \mathcal{T} e^{-i \int_{j\frac{t}{N}}^{(j+1)\frac{t}{N}} \alpha \tilde{H}_1^B(\tau) d\tau} \| \\
&\leq \alpha^2 \sum_{j=0}^{N-1} \int_{j\frac{t}{N}}^{(j+1)\frac{t}{N}} dv \int_{j\frac{t}{N}}^{v} ds \| [\tilde{H}_1^A(s), \tilde{H}_1^B(v)] \| = O\left(\frac{\alpha^2 t^2}{N}\right).
\end{aligned}
\quad (5)
$$

To turn this approach into a useful product-formula decomposition, we describe how to implement the time-ordered exponentials. This can be done using the definition of the time-ordered exponential in the other direction,

$$\mathcal{T} e^{-i \int_a^b d\tau \alpha \tilde{A}(\tau)} = e^{ibH_0} e^{-i(b-a)(H_0 + \alpha A)} e^{-iaH_0}, \quad (6)$$

which is valid for any Hermitian operator $\tilde{A}(t) = e^{iH_0 t} A e^{-iH_0 t}$. This leads to the decomposition

$$U_{\text{apx}} = e^{-itH_0} \mathcal{T} e^{-i \int_0^t \alpha \tilde{H}_1^A(\tau) d\tau} \mathcal{T} e^{-i \int_0^t \alpha \tilde{H}_1^B(\tau) d\tau} = e^{-it(H_0 + \alpha H_1^A)} e^{itH_0} e^{-it(H_0 + \alpha H_1^B)}. \quad (7)$$

This is nothing more than the usual first-order Trotter decomposition of the Hamiltonian $H = H_0 + \alpha(H_1^A + H_1^B)$ using the summands $H_0 + \alpha H_1^A$, $-H_0$, and $H_0 + \alpha H_1^B$.

The decomposition (7) has an error $\alpha$ times smaller than the usual first-order Trotter formula. In particular, we have the following theorem.

**Theorem 1.** (THRIFT decomposition) Given a Hamiltonian $H = H_0 + \alpha H_1$ where $H_1 = \sum_{\gamma=1}^{\Gamma} H_1^{\gamma}$, the decomposition

$$U_{\text{apx}}(t) := e^{-itH_0} \prod_{\gamma} \left( e^{itH_0} e^{-it(H_0 + \alpha H_1^{\gamma})} \right) \quad (8)$$

approximates $U(t) = e^{-itH}$ with error

$$\| U(t) - U_{\text{apx}}(t) \| \leq \alpha^2 \int_0^t dv \int_0^v ds \sum_{\gamma_1 < \gamma_2 = 1}^{\Gamma} \| [H_1^{\gamma_1}(s), H_1^{\gamma_2}(v)] \| . \quad (9)$$

For sufficiently small time, this error is $O(\alpha^2 t^2)$.

The proof of this theorem appears in "Methods". For $\alpha$ small the error of this approximation scales better than a normal Trotter approximation.

The THRIFT decomposition in Theorem 1 corresponds to a first-order Trotter formula, and can be used as a seed for higher-order approximations using standard techniques[1,12,25,26]. Note that, in practice, for this result to be useful, the unitary $e^{-it(H_0 + \alpha H_1^{\gamma})}$ has to be implemented with an error of order $O(\alpha^2)$. More formally, we have the following procedure to turn a product formula into a THRIFT formula with $O(\alpha^2)$ error scaling.

**Proposition 1.** (Higher-order THRIFT) Given a second-order product formula $\mathcal{S}_2(t)$ and a set of parameters $\{u_j\}_{j=1}^m$ such that

$$\mathcal{S}_k(t) = \prod_{j=1}^m \mathcal{S}_2(u_j t) \quad (10)$$

is a $k$th-order product formula, the product

$$\mathcal{S}_k(t) = \prod_{j=1}^m U_{\text{apx}}\left(\frac{u_j}{2} t\right) U_{\text{apx}}^\dagger\left(-\frac{u_j}{2} t\right), \quad (11)$$

with $U_{\text{apx}}(t)$ specified by Eq. (8), approximates $e^{-itH}$ with error $O(t^{k+1} \alpha^2)$.

**Proof.** $U_{\text{apx}}(t)$ is simply a first-order product formula with the unusual splitting

$$H = (H_0 + \alpha H_1^1) - H_0 + \cdots + (H_0 + \alpha H_1^\Gamma). \quad (12)$$

It follows trivially that Eq. (11) is a $k$th-order product formula. To prove the $O(\alpha^2)$ error scaling, we write

$$\mathcal{S}_k(t) = e^{-i \sum_j u_j t H_0} \prod_{j=1}^m \left( \prod_{\gamma=1}^{\Gamma} \mathcal{T} e^{-i \int_{(a_j + \frac{u_j}{2})t}^{(a_j + u_j)t} \alpha \tilde{H}_1^{\gamma}(\tau) d\tau} \right) \left( \prod_{\gamma=\Gamma}^{1} \mathcal{T} e^{-i \int_{a_j t}^{(a_j + \frac{u_j}{2})t} \alpha \tilde{H}_1^{\gamma}(\tau) d\tau} \right) \quad (13)$$

with $a_{m-k} = \sum_{r=0}^{k-1} u_{m-r}$ and $a_m = 0$. This expression follows by applying Eq. (6) to Eq. (11). A valid product formula satisfies $\sum_{j=1}^m u_j = 1$, so the

claim follows by Supplementary Theorem 3 of Supplementary Note 1. This finishes the proof.

In practice, the set of parameters $s := \{u_j\}_{j=1}^m$ in Proposition 1 can be taken from known results like the recursive definition $\mathcal{S}_{2k}(t) = \mathcal{S}_{2k-2}^2(v_k t)\mathcal{S}_{2k-2}((1-4v_k)t)\mathcal{S}_{2k-2}^2(v_k t)$, where $v_k = 1/(4-4^{1/(2k-1)})$ from[25]. For a sixth-order formula of this type, this means $s = \{(r,r,s,r,r),(r,r,s,r,r),(m,m,sm/r,m,m),(r,r,s,r,r),(r,r,s,r,r)\}$ with $r = v_2 v_4$, $s = (1-4v_2)v_4$, and $m = v_2(1-4v_4)$. In the numerical analysis that we perform later, we use the parameters found in ref. 26, see "Results".

## Beyond quadratic scaling

The procedure developed in Proposition 2 improves the $O(t)$ error scaling, but leaves the $O(\alpha^2)$ error scaling unchanged. In fact, in Supplementary Note 1 we prove that no formula that approximates the evolution by a product of time-ordered evolutions according to terms of the Hamiltonian can achieve better scaling in $\alpha$ than THRIFT, regardless of how the Hamiltonian is decomposed. However, in this section we show how to achieve better scaling using two alternative approaches.

Motivated by Eq. (2), we look for approximations of the time-ordered operator that have better error scaling in the small parameter $\alpha$. First, we consider the Magnus expansion[27], which approximates the time-ordered exponential as the standard exponential of a time-dependent operator $\Omega$. Second, we consider directly approximating the time-ordered exponential as a product of exponentials[28]. We show that these approaches achieve error scaling $O(t^k \alpha^k)$ for any positive integer $k$. We also present two algorithms to implement these approximations in practice. The first of these is consequence of the following result:

**Theorem 2.** (Magnus-THRIFT decomposition) Consider a Hamiltonian $H = H_0 + \alpha H_1$. Let $\tilde{H}_1(t) := e^{itH_0} H_1 e^{-itH_0}$. Defining $\Omega^{[k]} := \sum_{j=1}^k \Omega_j(\alpha, t)$, the operation

$$U_M(t) := e^{-itH_0} \exp\left(\Omega^{[k]}(\alpha, t)\right) = e^{-itH_0} \exp\left(\sum_{j=1}^k \alpha^j \tilde{\Omega}_j(t)\right) \quad (14)$$

approximates $U(t) = e^{-itH}$ with error $O((t\alpha)^{k+1})$ for small times $t$.

The proof of this theorem is presented in "Methods". We now describe a method for approximating the dynamics of the Hamiltonian $H = H_0 + \alpha H_1$ for time $T$ using the Magnus expansion, that we call Magnus-THRIFT Algorithm. The approach is as follows:

1. Write the evolution operator $U(T) = e^{-iT(H_0 + \alpha H_1)}$ in the interaction picture, with $H_0$ as the dominant part:

$$U(T) = e^{-iTH_0} \mathcal{T} e^{-i\int_0^T \alpha \tilde{H}_1(t)}. \quad (15)$$

2. Slice the time $T$ into $N$ intervals:

$$\mathcal{T} e^{-i\int_0^T \alpha \tilde{H}_1(t)} = \prod_{k=0}^{N-1} \mathcal{T} e^{-i\int_{k\frac{T}{N}}^{(k+1)\frac{T}{N}} \alpha \tilde{H}_1(t)}. \quad (16)$$

3. Approximate the time-ordered exponential of a slice using its Magnus expansion up to order $O((\frac{T}{N}\alpha)^p)$. Note that here we use the Magnus expansion with an initial time $t_0 \neq 0$. We write the Magnus approximation of order $p$ with an arbitrary initial time $t$ as $\Omega(\alpha, \delta t; t)$, such that

$$\mathcal{T} e^{-i\int_t^{t+\delta t} \alpha \tilde{H}_1(t)} = \exp\left(\Omega^{[p]}(\alpha, \delta t; t)\right) + O\left((\delta t \alpha)^{p+1}\right). \quad (17)$$

4. Approximate the exponential $\exp\left(\Omega^{[p]}(\alpha, \delta t; t)\right)$ obtained from the Magnus expansion using a $p$th-order product formula $S_p$:

$$\exp\left(\Omega^{[p]}(\alpha, \delta t; t)\right) = S_p(t, \delta t) + O\left((\delta t \alpha)^{p+1}\right). \quad (18)$$

This procedure leads to the decomposition

$$U(T) = e^{-iTH_0} \prod_{k=1}^N S_p\left((k-1)\frac{T}{N}, \frac{T}{N}\right) + O\left(N\left(\frac{T\alpha}{N}\right)^{p+1}\right). \quad (19)$$

Note that this last step does not alter the scaling with $\alpha$, as the norm of the time-dependent Hamiltonian that determines the time-evolution operator is $\alpha \|\tilde{H}_1(t)\|$, so the error has to have the same scaling in time and $\alpha$.

As an example, consider the expansion of

$$e^{\Omega^{[2]}(\alpha, t; \delta t)} = e^{-i\alpha\delta t\left(\frac{1}{\delta t}\int_t^{t+\delta t} d\tau H(\tau) - \frac{i\alpha}{2\delta t}\int_t^{t+\delta t} dt_1 \int_t^{t_1} dt_2 [H(t_1), H(t_2)]\right)}. \quad (20)$$

Expanding the time-dependent Hamiltonian as a sum of time-independent operators $O_q$ and functions of time $\alpha_q(t)$ as $H(t) = \sum_{q=1}^Q \alpha_q(t) O_q$, we find

$$\Omega^{[2]}(t, \delta t) = -i\alpha\delta t\left(\sum_q A_q(t, \delta t) O_q + \sum_{q>p} B_{qp}(t, \delta t)[O_q, O_p]\right) \quad (21)$$

where

$$A_q(t, \delta t) = \frac{1}{\delta t}\int_t^{t+\delta t} d\tau \alpha_q(\tau), \quad (22)$$

$$B_{qp}(t, \delta t) = -\frac{i\alpha}{4\delta t}\int_t^{t+\delta t}\int_t^{t+\delta t} dt_1 dt_2 \alpha_q(t_1)\alpha_p(t_2)\operatorname{sign}(t_1 - t_2), \quad (23)$$

which can be computed classically. Thus we can approximate $e^{\Omega^{[2]}(\alpha, t; \delta t)}$ using a second-order product formula as

$$
\begin{aligned}
e^{\Omega^{[2]}(\alpha, t; \delta t)} &= e^{-i\epsilon\delta t\left(\frac{1}{\delta t}\int_t^{t+\delta t} d\tau H(\tau) - \frac{i\epsilon}{2\delta t}\int_t^{t+\delta t} dt_1 \int_t^{t_1} dt_2 [H(t_1), H(t_2)]\right)} \\
&= e^{-i\epsilon\delta t\left(\sum_q A_q(t, \delta t) O_q + \sum_{q>p} B_{qp}(t, \delta t)[O_q, O_p]\right)} \\
&= e^{-\frac{i\epsilon\delta t}{2}\sum_q A_q(t, \delta t) O_q} e^{-i\epsilon\delta t \sum_{q>p} B_{qp}(t, \delta t)[O_q, O_p]} e^{-\frac{i\epsilon\delta t}{2}\sum_q A_q(t, \delta t) O_q} \\
&\quad + O(\alpha^3 \delta t^3).
\end{aligned}
$$

$$(24)$$

If necessary, each of the products can be decomposed further using a second-order product formula to keep the error at most $O(\alpha^3 \delta t^3)$.

Note that in any application of these formulas, some care has to be taken when expanding functions of time, to avoid losing the favourable scaling with $\alpha$. As the error scales with both $\alpha$ and $t$, in any expansion the scaling with both of them should be considered.

In principle it should be possible to analyse the commutator scaling of the product formula appearing in Eq. (19), generalizing[12]. We leave this as a topic for future work.

In Supplementary Note 2 we introduce another algorithm achieving $O(\alpha^{k+1} t^{k+1})$ error scaling. It is based on the Fer approximation[28] of the time-ordered operator in the interaction picture in Eq. (31), so we refer to it as Fer-THRIFT.

## Numerical simulations

The asymptotics derived in Theorems 1 and 3 show that for $\alpha$ small enough, THRIFT methods will outperform Trotter methods, and for

even smaller $\alpha$, Magnus-THRIFT will eventually outperform THRIFT. Similarly, higher-order methods will outperform lower-order methods for small enough time steps. To ascertain that THRIFT and Magnus-THRIFT methods give an advantage at relevant values of $\alpha$ and $T$, we performed extensive simulations of different models, namely the transverse-field Ising model in one and two dimensions, and the Heisenberg model with random local fields in one dimension (see below). Additionally, we present a similar analysis of a fermionic model, the 1D Fermi-Hubbard model, in Supplementary Note 4.

We compare the ordinary first- and second-order product formulas[1,12] (here dubbed "Trotter 1" and "Trotter 2"), the fourth-order formula due to Suzuki[25] (here dubbed "Trotter 4" for conciseness), a numerically optimised eighth-order product formula due to Morales et al.[26] ("optimised Trotter 8") based on an ansatz of Yoshida[29], and a fourth-order formula optimised for Hamiltonians containing a small perturbation derived in ref. [15] (here dubbed "opt. small $A$ 4" to indicate its error scaling with $T$; see Supplementary Note 3). For each of these product formulas, we also construct the corresponding THRIFT circuit (dubbed "THRIFT 1" through "THRIFT 4" and "optimised THRIFT 8") as described in Theorem 1 and Proposition 2. For the transverse-field Ising model, we also implement the Magnus-THRIFT decompositions described in Theorem 3 with the first- and second-order Magnus expansion.

In the numerical implementation of THRIFT 1 through 8, we use the approximant

$$\left(U_{\mathrm{apx}}(T/N)\right)^N = \left(e^{-i\frac{T}{N}H_0}\prod_\gamma\left(e^{i\frac{T}{N}H_0}e^{-i\frac{T}{N}(H_0+\alpha H_1^\gamma)}\right)\right)^N,$$

obtained by first breaking up the total time $T$ into small steps $T/N$ and then approximating each unitary evolution over a small step by Eq. (8). For a total time-independent Hamiltonian $H$, this is equivalent to splitting the time-ordered exponential over the full evolution time into a product of unitary evolutions with a small time step $T/N$, as described in Eqs. (15) and (16).

Note that Fer-THRIFT 1 and Magnus-THRIFT 1 coincide. As we found that Magnus-THRIFT 2 was not generally competitive with the other approaches for the systems we analysed, we did not implement Fer-THRIFT 2 as it has essentially the same cost as Magnus-THRIFT 2.

## 1D and 2D transverse-field Ising model with weak coupling

The first model we use for numerical tests and algorithm comparison is the transverse-field Ising model with weak interaction in one and two dimensions. In the 1D case, the model is integrable and can be mapped to a free-fermion model that can be simulated in polynomial time and space using the method described in refs. [30,31]. This enables us to simulate chains of length up to $L=100$ using the fermionic linear optics simulation tools from[32]. While the equivalence to free fermions makes this model a less interesting target for quantum simulation, we expect that the simulation costs may be indicative of costs for some other 1D models that are not necessarily classically easy. Indeed, we see evidence of this in the case of the Heisenberg model, as shown below. In 2D, we are restricted to relatively small system sizes using full state vector simulations.

The Hamiltonian of the transverse-field Ising model is

$$H_{\mathrm{TFIM}} = h\sum_j Z_j + J\sum_{\langle i,j\rangle} X_i X_j, \qquad (25)$$

where $X_i$ and $Z_i$ are the spin-1/2 operators in the $x$ and $z$ directions, respectively. For the purpose of studying THRIFT-based algorithms, we fix the field strength to $h=1$, let the interaction strength $\alpha := J$ be the small parameter, and measure time $T$ in units of $h^{-1}$. Since the

**Table 1 | Circuit depth comparison of the different TDS algorithms investigated and shown in Fig. 1 for the 1D transverse-field Ising model**

| Algorithm | 2-qubit gate depth | CNOT depth | # steps in Fig. 1 |
|---|---|---|---|
| Trotter 1 | $2N$ | $4N$ | 15 |
| Trotter 2 | $2N+1$ | $4N+2$ | 15 |
| Trotter 4 | $10N+1$ | $20N+2$ | 3 |
| optimised Trotter 8 | $30N+1$ | $60N+2$ | 1 |
| THRIFT 1 | $2N$ | $4N+2$ | 15 |
| THRIFT 2 | $2N+1$ | $4N+2$ | 15 |
| THRIFT 4 | $10N+1$ | $20N+2$ | 3 |
| optimised THRIFT 8 | $30N+1$ | $60N+2$ | 1 |
| Magnus-THRIFT 1 | $2N$ | $4N$ | 15 |
| Magnus-THRIFT 2 | $12N+3$ | $36N+9$ | 2 |
| optimised small $A$ 4 | $12N$ | $24N$ | 2 |

The first column shows the 2-qubit depth of the circuit corresponding to $N$ Trotter steps in terms of arbitrary 2-qubit gates. The second column shows the corresponding cost in terms of CNOT gates. Finally, the third column gives the number of Trotter steps used in Fig. 1, which correspond to a fixed budget of arbitrary 2-qubit gates of 31.

**Table 2 | Circuit depth comparison of the different TDS algorithms investigated and shown in Fig. 1 for the 2D transverse-field Ising model**

| Algorithm | 2-qubit gate depth | CNOT depth | # steps in Fig. 1 |
|---|---|---|---|
| Trotter 1 | $4N$ | $8N$ | 26 |
| Trotter 2 | $6N+1$ | $12N+2$ | 17 |
| Trotter 4 | $30N+1$ | $60N+2$ | 3 |
| optimised Trotter 8 | $90N+1$ | $180N+2$ | 1 |
| THRIFT 1 | $4N$ | $8N+2$ | 26 |
| THRIFT 2 | $6N+1$ | $12N+2$ | 17 |
| THRIFT 4 | $30N+1$ | $60N+2$ | 3 |
| optimised THRIFT 8 | $90N+1$ | $180N+2$ | 1 |
| Magnus-THRIFT 1 | $4N$ | $8N$ | 26 |
| Magnus-THRIFT 2 | $102N+3$ | $306N+9$ | 1 |
| optimised small $A$ 4 | $28N$ | $56N$ | 3 |

The first column shows the 2-qubit depth of the circuit corresponding to $N$ Trotter steps in terms of arbitrary 2-qubit gates. The second column shows the corresponding cost in terms of CNOT gates. Finally, the third column gives the number of Trotter steps used in Fig. 1, which correspond to a fixed budget of arbitrary 2-qubit gates of 105.

transverse-field part, $H_0 = \sum_j Z_j$, only consists of one-qubit terms, this has the advantage that the interaction-picture Hamiltonian $\tilde{H}_1(t)$ has the same locality as the original $H_1 = J\sum_{\langle i,j\rangle} X_i X_j$, and THRIFT circuits have the same 2-qubit gate depth as the corresponding Trotter circuits. We also note that, because $e^{-itJX_iX_j}$ and $e^{-it(JX_iX_j + h(Z_i+Z_j))}$ can be implemented with the same number of CNOT gates—namely two—the same holds for CNOT gate depth. The 2-qubit gate depths of one TDS step for all algorithms considered are shown in Tables 1 and 2 (2D). The explicit formulas for the approximants used for the THRIFT simulations of the transverse-field Ising model are discussed in Supplementary Note 3.

In Fig. 1, we show which of the different Trotter, THRIFT, or Magnus-THRIFT algorithms performs best at a given $T$ and $\alpha$ for a wide range of these two quantities for the 1D transverse-field Ising model (panel a) and 2D transverse-field Ising model (panel b). The results broadly agree with what we expect from Theorems 1 and 3 and Proposition 2: as $T$ decreases, higher-order formulas become advantageous over lower orders, and for smaller $\alpha$, THRIFT methods

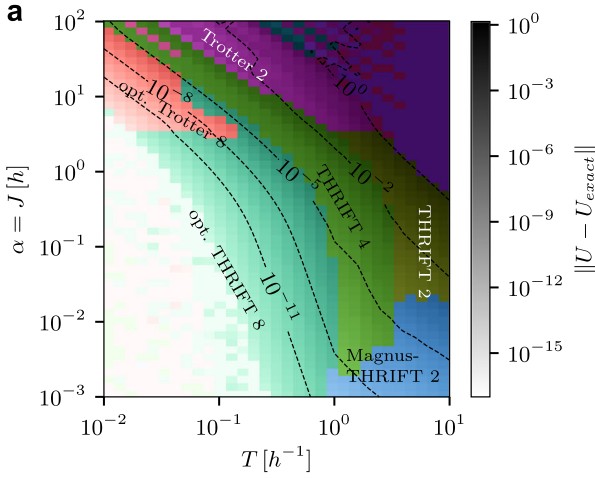
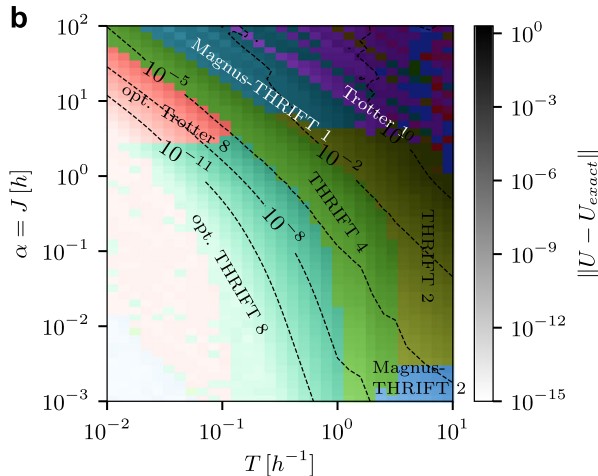

**Fig. 1 | Best time-dynamics simulation (TDS) algorithm for the 1D and 2D transverse-field Ising models (TFIM). a** Landscape of the best TDS algorithm, as measured by the worst-case error $\|U - U_{\text{exact}}\|$, as a function of the relative field strength $\alpha = J/h$ and evolution time $T$ at identical circuit depth for a $1 \times 16$ Ising chain with transverse field. The circuit depth was fixed to 1 step of Magnus-THRIFT 2 evolution. For the other algorithms, the number of steps is chosen to match the 2-qubit depth as closely as possible according to the 2-qubit depths shown in Tables 1 and 2. The colour of each point represents the algorithm that achieves the lowest error at those values of $J$ and $T$, while the brightness indicates the magnitude of the error. **b** Same for a $3 \times 3$ transverse-field Ising model. Note that in the top right corner of both panels, $\|U - U_{\text{exact}}\|$ is of order 1, so this region is not of particular interest.

are advantageous over Trotter methods. Interestingly, this crossover happens for a relatively large $\alpha \approx 3$ for the transverse-field Ising model. First-order methods are never advantageous for the 1D transverse-field Ising model, because for Hamiltonians that can be split into only two exactly implementable parts for Trotterisation, second-order methods have the same amortised depth per step as first-order methods (see Table 1). Magnus-THRIFT 2 outperforms all other methods only for very small $\alpha < 10^{-2}$ and $T > 1$. Nevertheless, in order to make a fair comparison between all algorithms we allowed for a gate budget corresponding to 15 and 26 steps of the first-order Trotter formula for the 1D and 2D case, respectively (see Table 1). This leads to errors below numerical accuracy for very small $\alpha$ and $T$.

An analysis to compare the algorithms at system sizes, evolution times, and worst-case errors relevant in near-term realistic implementations is done in Fig. 2. There we investigate the scaling of the different algorithms with the system size and evolution time by searching for the lowest number of steps such that each algorithm achieves worst-case error $\|U - U_{\text{exact}}\| \le 0.01$. For the 1D transverse-field Ising model, we scale the system size $L$ and evolution time $T$ together as $T = L$. Figure 2a shows the 2-qubit depth to get the error below threshold. For the 2D transverse-field Ising model, we fix the system size at $3 \times 3$ and only change the simulation time $T$ when searching for the minimal circuit depth to get the error below threshold. The results are shown in Fig. 2b. In both cases, we find that the circuit depth as a function of evolution time (and system size) is well described by a power law. The power-law exponents match those theoretically expected from Supplementary Note 1, with the notable exception of the optimised eighth-order THRIFT formula and fourth-order Trotter formula, for which the exponents are substantially smaller. In Supplementary Note 4 we show these exponents as a function of the interaction strength $J = \alpha$ and discuss the results in more detail. We observe surprisingly slow growth of the circuit depth for the optimised eighth-order THRIFT formula, which appears to scale sub-linearly in the evolution time. While the small slope of opt. THRIFT 8 for the 1D case could be attributed to the model being fast-forwardable, in the 2D case we believe this is an artifact of the small system size, as the model is not believed to be fast-forwardable in general. See Supplementary Note 4 for more details. The specific partitions we used to implement Trotter, Omelyan et al.'s, and THRIFT

algorithms for the various models we consider are discussed in Supplementary Note 3.

We now investigate the benefits of THRIFT methods in scenarios of more practical relevance. Specifically, in Fig. 3 we compare the simulation of the dynamics of an excitation placed in the ground state of a 29-site transverse-field Ising model chain with $J = 1$ and $h = 2$ (corresponding to $\alpha = J/h = 0.5$) obtained by using 15 steps of second-order Trotter and THRIFT formulas. Despite the two formulas requiring the same computational resources (see Table 1), one can clearly see that the simulation with second-order THRIFT remains close to the true dynamics for much longer than the simulation with the Trotter formula of the same order. The system sizes and circuit depths required to replicate Fig. 3 on real hardware are within the reach of current quantum devices[33]. Comparing the previous results for the 1D and 2D transverse-field Ising model, we expect the advantage of THRIFT methods over Trotter formulas observed here for the 1D transverse-field Ising model to extend to higher-dimensional cases as well, making THRIFT methods a powerful tool for the simulation of quantum spin models on near-term quantum devices in regimes of practical interest.

## 1D Heisenberg model with strong random fields

The second model we use for numerical tests of the THRIFT algorithms is the 1D spin-$\frac{1}{2}$ Heisenberg model with strong random fields. Unlike the 1D transverse-field Ising model, it is not exactly solvable, and we are not aware of a fast classical simulation for arbitrary times. The Hamiltonian is

$$H_{\text{Heisenberg}} = J \sum_{\langle i,j \rangle} \left( X_i X_j + Y_i Y_j + Z_i Z_j \right) + \sum_i h_i Z_i, \qquad (26)$$

where the $h_i$ are chosen uniformly random in $[-h, h]$ and $X_i$, $Y_i$, and $Z_i$ are again the spin-$\frac{1}{2}$ operators in the respective directions. We fix $h = 1$, use the interaction strength $\alpha := J$ as the small parameter, and measure time $T$ in units of $h^{-1}$. To evaluate errors, we always average over 10 different random instantiations of the field strengths $h_i$. As in the case of the transverse-field Ising model, the field part $H_0 = \sum_i h_i Z_i$ consists only of one-qubit terms, so $\tilde{H}_1(t)$ consists entirely of 2-qubit terms. Because simulating the Heisenberg interaction $e^{-it(X_i X_j + Y_i Y_j + Z_i Z_j)}$ already takes three CNOT gates, simulating the THRIFT gate

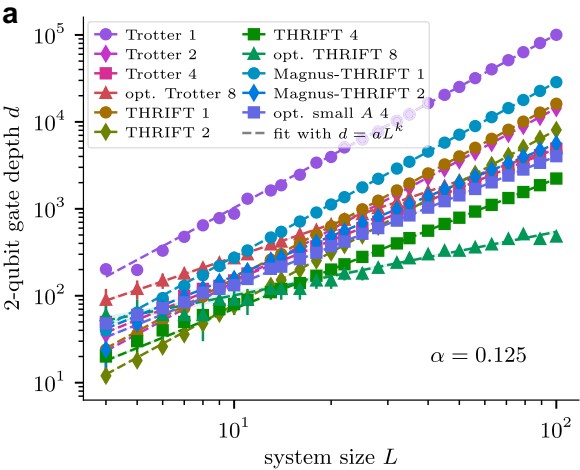
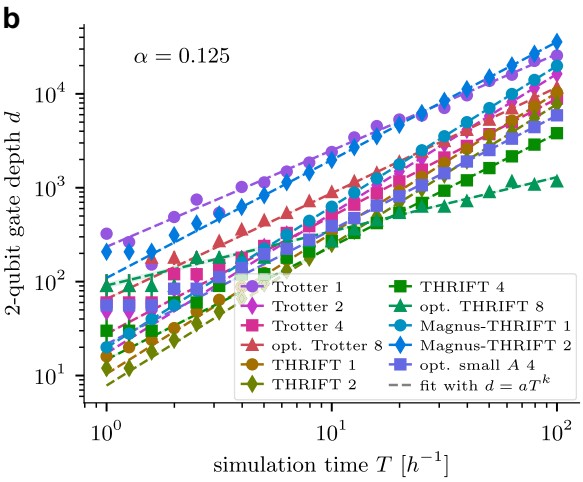

**Fig. 2 | 2-qubit gate depth scaling as a function of system size and simulation time for the 1D and 2D transverse-field Ising model (TFIM). a** 2-qubit gate depth to achieve $\|U - U_{\text{exact}}\| \leq 0.01$ for the different TDS algorithms for a field strength of $J = 1/8$ and evolution time $T = L$, for a $1 \times L$ Ising chain with transverse field. The depths follow a power law of the form $d = aL^k$ whose parameters $a$ and $k$ we determine via a least-squares fit and report, also for different values of $\alpha$, in

Supplementary Fig. 4. **b** Similar simulation for a $3 \times 3$ transverse-field Ising model. Because the 2D transverse-field Ising model is not integrable and hence large system sizes are not classically simulable, we fix the system size to $3 \times 3$ and only scale the evolution time $T$. Error bars (mostly barely visible) are $\pm 1$, i.e., the minimal possible depth resolution.

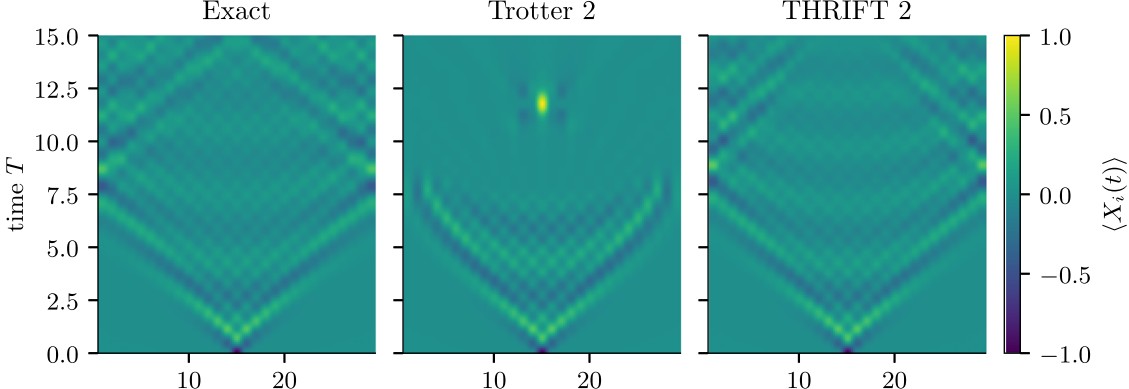

**Fig. 3 | Excitation dynamics using Trotter 2 and THRIFT 2.** Simulation of the propagation of an excitation in a 29-site transverse-field Ising model chain at $h = 2$ and $J = 1$ ($\alpha = 0.5$) with 15 Trotter 2/THRIFT 2 steps. The initial state chosen here is

$|\psi(0)\rangle = |\downarrow \cdots \downarrow - \downarrow \cdots \downarrow\rangle$, with $|-\rangle = (|\uparrow\rangle - |\downarrow\rangle)\sqrt{(2)}$. Clearly, the simulation with Trotter 2 starts to deviate significantly from the exact dynamics starting from $T > 5$, while the simulation with THRIFT 2 remains accurate for much longer times.

$e^{-it(X_iX_j + Y_iY_j + Z_iZ_j + h_iZ_i + h_jZ_j)}$ takes the same 2-qubit gate depth. Therefore, one step of any THRIFT circuit takes the same depth as one step of the corresponding Trotter circuit. See Supplementary Note 3 for more details about how we partitioned $H_{\text{Heisenberg}}$. The exact 2-qubit gate depths are shown in Table 3.

In Figs. 4 and 5, we repeat the analysis done for the transverse-field Ising model in Figs. 1 and 2 for the Heisenberg model. However, because the Heisenberg model is not integrable and average-case errors are much easier to compute than worst-case errors, we use the average infidelity as a figure of merit in Fig. 5. (Note that this may not be indicative of worst-case performance, since product formula simulations can have significantly better performance on average[34]). Similarly to the case of the transverse-field Ising model, the THRIFT methods perform better than the corresponding Trotter methods, with higher-order methods outperforming lower-order methods for smaller $T$ and $\alpha$ in Fig. 4. We observe that the crossover point from one method to the next in Fig. 4 roughly happens along lines of constant $\alpha T$. This is because the interaction-picture Hamiltonian $\tilde{H}_1(t)$ scales with $\alpha$, so the relevant scale for the Trotter errors is $\alpha T/N$. The seeming advantage of the optimised eigth-order formula at very small $\alpha T$ is for worst-case errors below the

numerical precision floor, so it is probably not borne out in reality. In Fig. 5, we see that the THRIFT methods always outperform the corresponding Trotter methods, and the 2-qubit gate depth to achieve average infidelity below a fixed threshold scales very similarly with $T$ and the system size $L$ for both methods, in broad agreement with the theory in Supplementary Note 1. Again, we note that the system size of $L = 100$, an average-case error $\leq 0.01$ and evolution time $T = 100$ are realistic targets for near-term simulations. Figure 5 can also be directly compared to Fig. 1 in ref. 34, which considers the same question (albeit only for Trotter and not for THRIFT methods) for the Heisenberg model at $J = 1$. That analysis finds very similar results, including matching exponents $k$. We present a more detailed analysis of the scaling of the circuit depth with system size and evolution time in Supplementary Note 2.

For this model, we did not implement the Magnus-THRIFT algorithm since we expect that it performs similarly to the 1D transverse-field Ising model case, i.e., it performs best only in a region with small $\alpha$ and large $T$. Furthermore, Magnus-THRIFT formulas of order $p > 1$ would involve unitaries acting on more than 2 qubits, resulting in a higher 2-qubit gate cost.

**Table 3 | Circuit depth comparison of the different TDS algorithms investigated and shown in Fig. 4 for the 1D Heisenberg model**

| Algorithm | 2-qubit gate depth | CNOT depth | # steps in Fig. 4 |
|---|---|---|---|
| Trotter 1 | $2N$ | $6N$ | 15 |
| Trotter 2 | $2N+1$ | $6N+3$ | 15 |
| Trotter 4 | $10N+1$ | $30N+3$ | 3 |
| optimised Trotter 8 | $30N+1$ | $90N+3$ | 1 |
| THRIFT 1 | $2N$ | $6N+3$ | 15 |
| THRIFT 2 | $2N+1$ | $6N+3$ | 15 |
| THRIFT 4 | $10N+1$ | $30N+3$ | 3 |
| optimised THRIFT 8 | $30N+1$ | $90N+3$ | 1 |
| optimised small $A$ 4 | $12N$ | $36N$ | 2 |

The first column shows the 2-qubit depth of the circuit corresponding to $N$ Trotter steps in terms of arbitrary 2-qubit gates. The second column shows the corresponding cost in terms of CNOT gates. Finally, the third column gives the number of Trotter steps used in Fig. 4, which correspond to a fixed budget of arbitrary 2-qubit gates of 31.

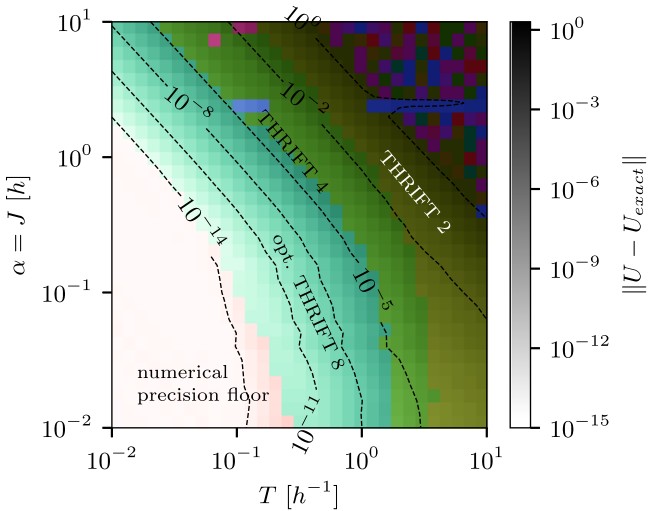

**Fig. 4 | Best time-dynamics simulation (TDS) algorithm for a 1D Heisenberg chain.** Landscape of the best TDS algorithm, as measured by the worst-case error $\|U - U_{\text{exact}}\|$, as a function of the relative field strength $\alpha = J/h$ and evolution time $T$ at identical circuit depth for a $1 \times 8$ Heisenberg chain. The circuit depth is fixed to one step of optimised THRIFT 8 evolution. For the other algorithms, the number of steps is chosen to match the 2-qubit depth as closely as possible according to the 2-qubit depths shown in Table 3. The colour of each point represents the algorithm that achieves the lowest error at those values of $J$ and $T$, while the brightness indicates the magnitude of the error. The isolated purple and red pixels in the THRIFT 4 and THRIFT 2 regions are artifacts of the randomness in the field strengths and running the optimised small $A$ and eighth-order simulations with different random fields, but do not seem indicative of the general relative performance of the algorithms at these $(\alpha, T)$-points.

## Discussion

Better algorithms to simulate the time dynamics of Hamiltonians with different scales have natural applications in systems where the interactions have distinct origins. We have shown both theoretically and through numerical experiments in various systems that the THRIFT algorithms can achieve better scaling than standard product formulas for Hamiltonians with different energy scales. Concretely, we consider Hamiltonians of the form $H = H_0 + \alpha H_1$, where $\alpha \ll 1$ and the norms of $H_0$ and $H_1$ are comparable. Using product formulas with a carefully chosen partition, we can achieve an $O(\alpha^2 t^k)$ error scaling for any $k \in \mathbb{N}$, which

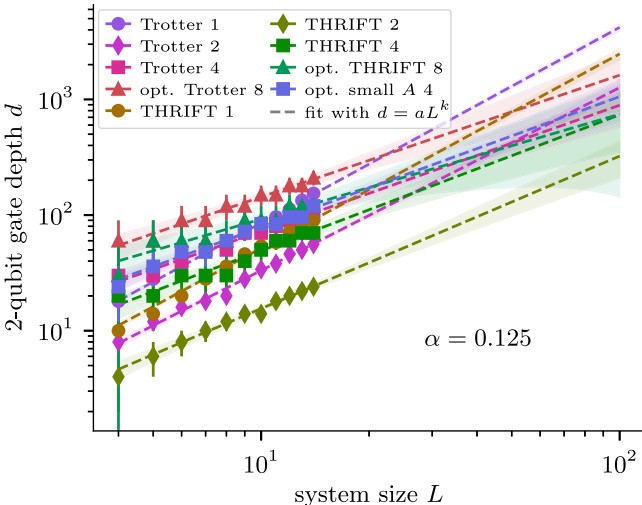

**Fig. 5 | 2-qubit depth scaling as a function of the Heisenberg chain size.** 2-qubit depth to achieve average infidelity $\mathbb{E}_{\{|x\rangle\}}[1 - |\langle x|U_{\text{exact}}^\dagger U|x\rangle|^2] \le 0.01$ for the different TDS algorithms for a $1 \times L$ Heisenberg chain with field strength of $J = 1/8$ and evolution time $T = L$. Unlike Fig. 2, we use average fidelity to be able to simulate larger systems. Again, the required depths follow a power law of the form $d = aL^k$ whose parameters $a$ and $k$ we determine via a least-squares fit and use to extrapolate to up to $L = 100$. We report the fit parameters $a$ and $k$, also for different values of $\alpha$, in Supplementary Fig. 5. Error bars are $\pm 1$ step and the shaded regions are the one-sigma confidence intervals of the extrapolations.

is better by a factor of $\alpha$ compared to the standard product formulas that do not use any structure of the Hamiltonian. We also present two algorithms to achieve scaling $O(\alpha^k t^k)$ of the approximation error. These two algorithms perform better than other formulas only in small, extreme regions of the parameter space of the systems we consider. However, such a scaling with $\alpha$ cannot be achieved using products of time-ordered evolutions according to the terms of the Hamiltonian, and they may achieve better performance in other applications.

While we have concentrated on the evolution generated by time-independent Hamiltonians, the methods developed in this work also generalise to time-dependent Hamiltonians satisfying the same assumptions. Consider a Hamiltonian $H(t) = H_0(t) + \alpha H_1(t)$, where $H_0(t)$ and $H_1(t)$ are time dependent and have similar norms for all times $t$. As before we consider $\alpha$ small. Using the same ideas developed in Results, it is possible to show that for a partition of $H_1(t) = H_1^A(t) + H_1^B(t)$, evolving the system with the approximant

$$U_{\text{apx}}(t, 0) := \mathcal{T}e^{-i\int_0^t H_0(s)ds}\mathcal{T}e^{-i\int_0^t \tilde{H}_1^A(s)ds}\mathcal{T}e^{-i\int_0^t \tilde{H}_1^B(s)ds}, \quad (27)$$

induces an error bounded by

$$\| \mathcal{T}e^{-i\int_0^t H(s)ds} - U_{\text{apx}}(t, 0) \| \le \alpha^2 \int_0^t dv \int_0^v ds \, \| [\tilde{H}_1^A(s), \tilde{H}_1^B(v)] \| , \quad (28)$$

where $\tilde{H}_1^{A,B}(t) := \mathcal{T}e^{i\int_0^t H_0(s)ds}H_1^{A,B}(t)\mathcal{T}e^{-i\int_0^t H(s)ds}$. The main difference with respect to the time-independent case is that the evolution over a total time $T$ cannot generically be obtained from repeating the evolution over small times, but instead must be obtained from an approximation of each time-ordered slice of the total evolution.

Although these algorithms lack the competitive scaling of other approaches not based on product formulas, it has been shown[10] that in the regime of medium sizes and time evolution scaling with the system

size, standard product formulas can outperform asymptotically better algorithms. This makes our approach competitive in practical applications.

Developing algorithms that utilise the structure of the Hamiltonian to lower the cost of simulating time dynamics is crucial to make quantum computers useful sooner. In particular, our approach may help to study dynamical phase transitions[35], where the behaviour of the dynamics of a system can change as a function of the parameters of the Hamiltonian. Quantum algorithms for time dynamics that fare well in particular regions of the parameter space allow exploring these questions with fewer resources, or for longer times given fixed resources and error.

## Methods

In this section we present the proof of Theorems 1 and 3 of the main text.

*Proof of Theorem 1.* Define the approximant

$$\| R_k(\Omega(\alpha, t)) \| \le \sum_{n=k+1}^{\infty} \alpha^n \| \tilde{\Omega}_n(t) \| \quad \text{using the triangle inequality and the definition of the remainder}$$

$$\le \frac{1}{2} \sum_{n=k+1}^{\infty} \frac{\alpha^n}{n!} \frac{d^n}{dz^n}(G^{-1}(0)) \left( 2 \int_0^t \| \tilde{H}_1(x) \| dx \right)^n \quad \text{applying Lemma 4 termwise} \tag{34}$$

$$= R_k\left( \frac{1}{2} G^{-1}\left( 2\alpha \int_0^t \| \tilde{H}_1(s) \| ds \right) \right) \quad \text{using the definition of the remainder .}$$

$$V^{(j)}(t) := \left( \prod_{k=1}^{j} \mathcal{T} e^{-i \int_0^t \tilde{H}_1^k(s) ds} \right) \mathcal{T} e^{-i \int_0^t \sum_{k=j+1}^{\Gamma} \tilde{H}_1^k(s) ds}. \tag{29}$$

Here $V^{(0)}(t) = \mathcal{T} e^{-i \int_0^t \tilde{H}_1(s) ds}$ corresponds to the evolution under the full Hamiltonian $\tilde{H}_1(t)$, while $V^{(\Gamma-1)}(t) = e^{itH_0} U_{apx}(t)$, where $U_{apx}(t)$ is defined in Eq. (8). This follows from repeated use of Eq. (6). Using the invariance of the operator norm and Eq. (4), it follows that

$$\| V^{(j)}(t) - V^{(j+1)}(t) \| \le \alpha^2 \int_0^t dv \int_0^v ds \sum_{k=j+2}^{\Gamma} \| [\tilde{H}_1^{j+1}(s), \tilde{H}_1^k(v)] \|. \tag{30}$$

We use Eq. (30) to bound the error by applying the triangle inequality on the identity $V^{(0)} - V^{(\Gamma-1)} = \sum_{j=0}^{\Gamma-2}(V^{(j)} - V^{(j+1)})$ and noting that $\|V^{(0)}(t) - V^{(\Gamma-1)}(t)\| = \|U(t) - U_{apx}(t)\|$, which leads finally to Eq. (9) as claimed. This finishes the proof.

In order to prove Theorem 3, it is convenient to introduce

$$\mathcal{T} e^{-i\alpha \int_0^t \tilde{H}(s) ds} =: e^{\Omega(\alpha, t)} \tag{31}$$

for some time-dependent operator $\Omega(\alpha, t)$, it is easy to show that $\frac{de^{\Omega(t)}}{dt} e^{-\Omega(t)} = -i\alpha \tilde{H}_1(t)$. Magnus[27] used this to find an equation for $\Omega$ by employing the inverse of the derivative of the exponential map, i.e.,

$$\frac{de^{\Omega(t)}}{dt} e^{-\Omega(t)} = \frac{e^{\mathrm{ad}_\Omega} - 1}{\mathrm{ad}_\Omega} \frac{d\Omega}{dt} \quad \rightarrow \quad \frac{d\Omega}{dt} = \frac{\mathrm{ad}_\Omega}{e^{\mathrm{ad}_\Omega} - 1}(-i\alpha H_1)$$

$$= \sum_{k=0}^{\infty} \frac{b_k}{k!} \mathrm{ad}_\Omega^k(-i\alpha H_1), \tag{32}$$

where $\mathrm{ad}_\Omega(\cdot) := [\Omega, \cdot]$ and $\mathrm{ad}_\Omega^j(\cdot) := \mathrm{ad}_\Omega^{j-1}([\Omega, \cdot])$. The coefficients $b_j$ are Bernoulli numbers, defined through $\frac{x}{e^x-1} = \sum_{j=0}^{\infty} \frac{b_j}{j!} x^j$. The equation for

$\Omega$ can now be solved through Picard iteration[27,36]. Defining $\alpha$-independent coefficients $\tilde{\Omega}_j(t)$ so that $e^{\Omega(\alpha,t)} = \exp\left( \sum_{j=1}^{\infty} \alpha^j \tilde{\Omega}_j(t) \right)$, and using this expression in Eq. (32), produces the recurrence[37]

$$\frac{d}{dt} \tilde{\Omega}_n(t) = \sum_{k=1}^{n-1} \frac{b_k}{k!} \sum_{\substack{j_1+j_2+\cdots+j_k=n-1 \\ j_1, j_2, \ldots, j_k \ge 1}} [\tilde{\Omega}_{j_1}(t), [\tilde{\Omega}_{j_2}(t), \ldots [\tilde{\Omega}_{j_k}(t), -i\tilde{H}_1(t)]\ldots]]. \tag{33}$$

The series for $\Omega$ converges for sufficiently small time $t$[38,39] (see also Supplementary Theorem 11). Using these results, we can state the following lemma bounding the terms of the Magnus expansion.

**Lemma 1.** For $l \ge 1$, $\|\tilde{\Omega}_l(t)\| \le \frac{1}{2} x_l (2 \int_0^t \|\tilde{H}_1(s)\| ds)^l$, where $x_l$ is the coefficient of $s^l$ in the expansion of $G^{-1}(s) = \sum_{m=1}^{\infty} x_m s^m$, the inverse function of $G(s) = \int_0^s (2 + \frac{x}{2}(1 - \cot(x/2))^{-1} dx$.

This lemma is mentioned in ref. 36. We include a proof for completeness in Supplementary Note 2. Armed with Lemma 4, we have:

*Proof of Theorem 3.* As $e^{-it(H_0 + \alpha H_1)} = e^{-itH_0} \mathcal{T} e^{-i\alpha \int_0^t \tilde{H}_1(s)}$, it suffices to approximate the time-ordered evolution $\mathcal{U}(\alpha, t) := \mathcal{T} e^{-i\alpha \int_0^t \tilde{H}_1(s)}$.

Introducing the Taylor remainder of a function $h(\alpha)$ as $R_k(h(\alpha)) := \sum_{n=k+1}^{\infty} \frac{\alpha^n}{n!} h^{(n)}(0)$, it follows that for $\Omega(\alpha, t) = \sum_{j=1}^{\infty} \alpha^j \tilde{\Omega}_j(t)$, one has $R_k(\Omega(\alpha, t)) = \Omega(\alpha, t) - \Omega^{[k]}(\alpha, t)$, and

The remainder provides a bound on the difference between $\mathcal{U}(\alpha, t) = e^{\Omega(\alpha, t)} = e^{\left( \Omega^{[k]}(\alpha, t) + R_k(\Omega(\alpha, t)) \right)}$ and $e^{\Omega^{[k]}(\alpha, t)}$ by means of the integral representation of the error

$$F := e^{\Omega(\alpha, t)} e^{-\Omega^{[k]}(\alpha, t)} - 1 = \int_0^1 ds e^{s(\Omega^{[k]}(\alpha, t) + R_k(\alpha, t))} R_k(\alpha, t) e^{-s\Omega^{[k]}(\alpha, t)}. \tag{35}$$

Using Eq. (34), we have $\| \mathcal{U}(\alpha, t) - e^{\Omega^{[k]}} \| \le R_k(\frac{1}{2} G^{-1}(\alpha t \| H_1 \|))$. This implies that the error scales as $O((\alpha t)^{k+1})$. This finishes the proof. Note that this result extends trivially to an arbitrary time-dependent $\tilde{H}_1(t)$.

## Data availability

Data supporting the figures and tables in this manuscript are available at ref. 40.

## Code availability

Numerical simulations have been carried out using the Yao.jl framework. The code for data analysis is provided in ref. 40.

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

## Acknowledgements

We thank J. Ostmeyer for pointing out ref. 15. This work received funding from the European Research Council (ERC) under the European Union's Horizon 2020 research and innovation programme (grant agreement No. 817581 (A.M.)), and from EPSRC grant EP/S516090/1 (A.M.), InnovateUK grant 44167 (A.M.), and InnovateUK grant 10032332 (A.M.). Andrew Childs's contribution to this publication was not part of his University of Maryland duties or responsibilities.

## Author contributions

R.S. initiated the project and designed the THRIFT algorithms. J.L.B. and F.M.G. implemented the algorithms for the specific models and performed the numerical simulations and analysis. C.D. performed the error scaling analysis of the Magnus-THRIFT algorithm. A.M.C. and A.M. provided technical guidance. All authors contributed to drafting and reviewing the paper.

## Competing interests

A.M. is a cofounder of Phasecraft Ltd. A patent application has been filed based on this work by R.S. with reference number 24158476.2 (EU). The remaining authors declare no competing interests.
