## [Transparent Peer Review file · Nature Communications]

Efficient and practical Hamiltonian simulation from time-dependent product formulas

Corresponding Author: Dr Raul Santos

Version 0:

Reviewer comments:

Reviewer #1

(Remarks to the Author)

In this work, the authors propose a suite of new algorithms to simulate time-independent Hamiltonian dynamics, optimized for the scenario where an initial, easy-to-simulate Hamiltonian, is perturbed by some interaction. The key ingredient in the algorithms they present is a novel use of simulating the time-independent dynamics using the interaction picture. More specifically, they observe that a specific product formula splitting of the terms that make up the perturbation allows one to bound the error of the simulation with a higher power of α where $\alpha \ll 1$ is the strength of the perturbation. The larger power of α within the simulation error naturally translates into better time complexity for the simulation when compared to the naive approach. They present a first-order product formula algorithm that exploits this observation before generalizing it to the k th-order product formula approach. They also prove no product formula approach can have a simulation error scaling better than $O(\alpha^2)$ and in order to go beyond this quadratic scaling, they present algorithms based on the Magnus expansion and Fer exponential approximations to the time-order exponential of the interaction Hamiltonian. Both these post-product formula algorithms leverage the same unique splitting of the perturbation Hamiltonian used in the product formula approaches, and similarly, they find an improved scaling of the simulation error with respect to α . Finally, they test out their algorithms numerically against other product formula approaches on the 1D/2D Transverse Field Ising model, the 1D Fermi-Hubbard model, and the 1D Heisenberg model, finding the expected results that their algorithms perform best for smaller values of the perturbation and longer times.

Developing resource-efficient quantum simulation algorithms can be highly impactful and certainly of interest to the broad community of physicists. I find that the algorithms and techniques developed in this work are very interesting and potentially useful. I am inclined to recommend this manuscript for publication, but there are some questions and comments that need to be addressed.

- Given the relevance of Refs 7 and 18, I do think a more quantitative comparison between the scaling of the algorithms seems to be necessary. This might require an analysis of the commutator scaling of the product formula appearing in Eq. (26).

- I believe the authors not only assume that H_0 can be efficiently simulated without any scaling dependence on time, but also that $H_0 + \alpha H_1^{(i)}$ can be efficiently simulated as well. Although this assumption does not contradict any of the final results since one can choose to partition the perturbation H_1 into $\{H_1^{(i)}\}$ such that each $H_0 + \alpha H_1^{(i)}$ can also be efficiently simulated, this assumption should still be explicitly mentioned. If one does not assume the above, then a naive attempt to Trotterize $H_0 + \alpha H_1^{(i)}$ into $\{H_0, \alpha H_1^{(i)}\}$ could reduce the overall scaling of the total error of simulation from $O(\alpha^2)$ to $O(\alpha)$. If my understanding is correct, this important assumption needs to be clearly stated where it is appropriate (including the introduction) and also justification on when this setup is relevant would be useful.

- I suggest authors provide more discussion around the various steps in the algorithm descriptions of both Magnus/Fer-THRIFT. More specifically, one point I feel needed to be addressed, was how in step 4 of both algorithms, the use of a p th order product formula does not reduce the scaling of the overall error with respect to α from beyond quadratic to quadratic. Although a closer inspection confirms that step 4 of both algorithms does not incur any penalty on the scaling of the error with respect to α , I do feel providing some intuition or a brief discussion on why this is not the case could be

helpful to readers.

- In proof of Theorem 4: I believe the left-hand-side of equation (20) was supposed to be $\|\Omega(\alpha, t) - \Omega^{[k]}(\alpha, t)\|$, reflecting the difference between the truncated Magnus series $\Omega^{[k]}(\alpha, t)$, and the complete series $\Omega(\alpha, t)$. I am not sure why $\|R_k(\Omega(\alpha, t))\|$ appears on the left-hand-side of equation (20) instead. Similarly, I believe the bounds on the integral in equation (21) need to range from 0 to 1 instead of 0 to s . Changing the upper bound from s to 1 makes the left-hand side of equation (22) consistent with the right-hand side.

- I believe the proof of Appendix A.3 is based on the assumption that $\{a_v\}$ are independent of α . Is that a necessary assumption? Is relaxing this assumption meaningful? and does it provide to avoid the nogo?

- I found using $H_{-1}(t)$ to denote H_{-1} in the interaction picture a bit confusing (below eq 2 and after). Maybe using a tilde?

Reviewer #2

(Remarks to the Author)

Reviewer #3

(Remarks to the Author)

I have read the manuscript with interest. Product formulas are central to my work and advancements in their performance and scope are important for quantum computing (and beyond).

My understanding of the main result is that it is a new product formula construction particularly well suited to the simulation of Hamiltonians of the form $H = H_0 + \alpha H_1$, with α small. The authors refer to the method as THRIFT. (The acronym is a bit forced but I'll give it a pass).

While the construction is obtained from an interaction picture representation -- which requires time evolution under time-dependent Hamiltonians -- the result is a specific product formula for time-independent Hamiltonian simulation, namely that of Eq. (8). This product formula includes a product over easy-to-implement exponentials from the decomposition of H_1 , as well as direct exponentials over H_0 . The exponential over H_0 will also need to be decomposed as a product formula, except for the special cases where it is already fast-forwardable (as in the examples they consider). They also demonstrate that the construction works for high-order formulas.

The appeal of this construction is that we obtain an extra factor of α in the error compared to more standard methods like the Suzuki hierarchy. This can be significant enough in practice to motivate the use of this method. The results are then expanded using techniques to decompose the time-ordered exponential. The authors refer to these as Magnus-THRIFT (based on the Magnus operator) and Fer-THRIFT (based on results by Fer to express time-ordered exponential as a product formula). These methods are more complicated, but achieve even better scaling with α for higher orders.

The paper concludes with numerical experiments on simple spin models to evaluate regimes where each of these methods is most appealing. While the numerical results are a bit challenging to parse, my general impression is that the techniques from this work become appealing whenever long time evolutions with extremely low error are required.

Overall I find that this paper introduces a useful result for the field that is well presented and, to the best of my ability, all derivations and theorems are correct. I will consider these results in my work if applications arise that meet the conditions where these methods become attractive.

Having said that, unfortunately my assessment is that Nature Communications is not the right fit for this work. Instead, I view this work as a "rapid communication": a clever and useful technique to keep in the arsenal for people working on applied quantum computing. In fact, my main criticism of the manuscript is that it contains too much information that dilutes the main results. Pages 1-5 contain the bulk of important results. The Magnus and Fer extensions could be cited as results, with derivations left for an appendix. In particular, I find that the numerical experiments don't add much extra value (and I generally love numerics!). One key numerical study for a representative system that draws clearer conclusions about the regime of applicability would have likely sufficed. I'm not recommending that the authors re-write the manuscript this way, but rather use this example to make the point that core of results and contributions don't evidently meet the standards of Nature Communications, and could otherwise find a better home in a different journal.

The following are more minor points that I share with authors in case they find them useful:

1. I'm not sure about the title. It seems to focus more on *how* the main result is derived rather than on describing the main result -- product formulas with lower error for "near-integrable" Hamiltonians.

2. It would be helpful to draw clearer conclusions on the regimes where this method is most appealing. Even better, it would strengthen the results significantly to showcase that this product formula approach significantly reduces cost of

implementation for applications of practical importance. (I'm wondering about this myself. Not sure I can identify where these ideas would have the strongest impact).

3. It's not so clear where the inequality in Eq. (4) comes from. Perhaps there is a succinct way to help the reader follow the derivation better.

4. In Prop. 2, how are the parameters u_j chosen in practice? This was not very clear.

5. The numerics sections consider very small errors, below $1e-6$ and all the way down to $1e-15$. In my experience this is overkill in practice. Said differently, applications where extremely low errors are needed are not typically where it becomes appealing to use product formulas compared to other algorithms for Hamiltonian simulation (and even less clear that quantum computers are attractive since this implies similarly low error rates). It would strengthen the results to better justify why such low errors are considered and meaningful

Version 1:

Reviewer comments:

Reviewer #1

(Remarks to the Author)

Overall the latest draft of "Efficient and practical Hamiltonian simulation from time-dependent product formulas" has addressed most of our comments raised in our first review, as well as added content which has improved the quality of the latest version. Authors consider our suggestion to include analytical comparison to the algorithms of references [7, 18] outside scope of this paper, which we disagree with. Although we think such analysis would make the work much stronger, given the importance and quality of the work, we still recommend the publication of the work in Nature Communications.

Re-stating our previous points.

1- We asked the authors to provide a more analytical comparison to the algorithms of references [7, 18]. This would most likely entail an analysis of the commutator error terms which show-up due to them employing product formulas as subroutines for both Magnus and Fer-THRIFT.

2- We asked the authors to make evident the unspoken assumption that the Hamiltonian $H_{\{0\}} + \alpha H_{\{1\}}^{(i)}$ is easy to simulate as well as $H_{\{0\}}$,

3- We requested them to clear-up the ambiguity about whether using product formulas as a subroutine in the Magnus/Fer-THRIFT algorithms reduces the scaling of the overall error on α .

4- In addition, we found two typos in the proof of Theorem 4.

5- We also asked them to discuss how making the intervals of time $[a_{\{v-1\}t}, a_{\{v\}t}]$ dependent on α effects their proof of no-go for product formulas with error scaling $O(\alpha^p)$ with $p > 2$.

6- Finally we asked them to make it more clear when the interaction picture Hamiltonian is being used and when it is not. We suggested they tilde the interaction picture Hamiltonian.

Point 2.) was addressed on both pages 2 and 5, with the new draft making this assumption much more evident by stating it in the introduction as well as when it is specifically applied. Point 3.) was addressed on page 7, and they also correct the two typos found in the proof of Theorem 4 on page 6. Point 5.) is addressed in a footnote on page 18, which makes more clear why α dependence of $[a_{\{v-1\}t}, a_{\{v\}t}]$ does not add or subtract from their result since one is allowed the freedom to choose the best set $\{a_{\{v\}}\}$. Finally they address point number 6.) by using $\tilde{H}(t)$ to represent the interaction Hamiltonian throughout the paper.

Only point 1.) was not addressed in the revised draft. Instead, the authors argue that analyzing how the commutators scale in-order to analytically compare their algorithm to that of references [7, 18] is complex and out of the scope of their work. We think lack of scaling analysis (even if not tight) would make the comparison with [7, 18] impossible, at least beyond the limited numerical experiments presented. The added numerical simulations Transverse Field Ising model is certainly helpful.

In-addition to addressing these points, the authors also moved the Fer-THRIFT algorithm, and the numerical analysis on the 1D Fermi-Hubbard model to the appendices. They also added an additional numerical analysis where they simulate a 29-site 1D Transverse Field Ising model and provide a heat-map of the 1-body $\langle X \rangle$ expectation values over time. They then compare the heat-map of the exact simulation to the heat-map of the 2nd-order Trotter and 2nd-order THRIFT simulations. These changes make the main draft more compact and provide an easy visualization of how their algorithm is able to stay more accurate to the exact simulation than generic 2nd-order Trotter over longer periods of time.

Two additional issues with this current version of the draft.

1- In equation (66), there is a comma after $\pi_{\nu} \gamma_{\{1\}}$ which is typo.

2- In equation (67-68), we believe the upperbound to the inner integral should be $a_{\{v-1\}}$ and not $a_{\{v\}}$.

Reviewer #2

(Remarks to the Author)

Response to reviewers

Below, we provide detailed replies to reviewers' reports, which are included for convenience in italic and black fonts, while our replies are in blue. Moreover, in the resubmission we are also including a pdf file in which the main changes with respect to the previous version are highlighted in blue.

Reviewer #1 (Remarks to the Author):

In this work, the authors propose a suite of new algorithms to simulate time-independent Hamiltonian dynamics, optimized for the scenario where an initial, easy-to-simulate Hamiltonian, is perturbed by some interaction. The key ingredient in the algorithms they present is a novel use of simulating the time-independent dynamics using the interaction picture. More specifically, they observe that a specific product formula splitting of the terms that make up the perturbation allows one to bound the error of the simulation with a higher power of α where $\alpha \ll 1$ is the strength of the perturbation. The larger power of α within the simulation error naturally translates into better time complexity for the simulation when compared to the naive approach. They present a first-order product formula algorithm that exploits this observation before generalizing it to the k th-order product formula approach. They also prove no product formula approach can have a simulation error scaling better than $O(\alpha^2)$ and in order to go beyond this quadratic scaling, they present algorithms based on the Magnus expansion and Fer exponential approximations to the time-order exponential of the interaction Hamiltonian. Both these post-product formula algorithms leverage the same unique splitting of the perturbation Hamiltonian used in the product formula approaches, and similarly, they find an improved scaling of the simulation error with respect to α . Finally, they test out their algorithms numerically against other product formula approaches on the 1D/2D Transverse Field Ising model, the 1D Fermi-Hubbard model, and the 1D Heisenberg model, finding the expected results that their algorithms perform best for smaller values of the perturbation and longer times.

Developing resource-efficient quantum simulation algorithms can be highly impactful and certainly of interest to the broad community of physicists. I find that the algorithms and techniques developed in this work are very interesting and potentially useful. I am inclined to recommend this manuscript for publication, but there are some questions and comments that need to be addressed.

We would like to thank the Reviewer for the positive assessment of our manuscript, for recommending publication, and for the insightful comments. In what follows we will provide a point-by-point response to Reviewer's comments and questions.

- Given the relevance of Refs 7 and 18, I do think a more quantitative comparison between the scaling of the algorithms seems to be necessary. This might require an analysis of the commutator scaling of the product formula appearing in Eq. (26).

We thank the Reviewer for this suggestion, but respectfully disagree. The analysis of commutator scaling of time-dependent formulas is an active area of research. e.g., in Mizuta et al., arXiv:2410.14243, an analysis of this type can be found, where the error is not only due to commutators but also arises from the derivatives of the Hamiltonian. Applied to our context, this would generate a full commutator scaling of our formulas even for the time-dependent part. We think though that this analysis is out of scope for the current paper, as the error bounds are loose and an ultimate comparison between the different algorithms is done numerically.

- I believe the authors not only assume that $H_{\{0\}}$ can be efficiently simulated without any scaling dependence on time, but also that $H_{\{0\}} + \alpha H_{\{1\}^{(i)}}$ can be efficiently simulated as well. Although this assumption does not contradict any of the final results since one can choose to partition the perturbation $H_{\{1\}}$ into $\{H_{\{1\}^{(i)}}\}$ such that each $H_{\{0\}} + \alpha H_{\{1\}^{(i)}}$ can also be efficiently simulated, this assumption should still be explicitly mentioned. If one does not assume the above, then a naive attempt to Trotterize $H_{\{0\}} + \alpha H_{\{1\}^{(i)}}$ into $\{H_{\{0\}}, \alpha H_{\{1\}^{(i)}}\}$ could reduce the overall scaling of the total error of simulation from $O(\alpha^2)$ to $O(\alpha)$. If my understanding is correct, this important assumption needs to be clearly stated where it is appropriate (including the introduction) and also justification on when this setup is relevant would be useful.

We agree with the Reviewer and indeed the possibility to implement efficiently the time evolution operator associated with each partition $H_{\{0\}} + \alpha H_{\{1\}^{(i)}}$ is an important requirement for the efficiency of the product formulas with introduced in our work. In the new version of the manuscript we have made this point more explicit in both the introduction and in the derivation of our formulas. We would like to point out that if $H_{\{0\}}$ consists of 1-local terms, we expect our formulas to outperform standard formulas. In this case, indeed, the cost to implement each of the terms $e^{-iH_{\{1\}^{(i)}}t}$ and $e^{-i(H_{\{0\}} + \alpha H_{\{1\}^{(i)}})t}$ would be similar. We also expect our formula to be competitive even in the case in which $H_{\{0\}}$ and $H_{\{1\}^{(i)}}$ are k -local Hamiltonians with a similar interaction graph. As the Reviewer pointed out, the key is for the cost to implement $e^{-i(H_{\{0\}} + \alpha H_{\{1\}^{(i)}})t}$ to not be significantly larger than the cost to implement $e^{-iH_{\{1\}^{(i)}}t}$ alone. This happens, for example, for the Fermi-Hubbard model, where we showed that indeed our formulas outperform standard ones only in a small region of the parameter space when the cost of implementing the gates is included. We would like to highlight that our approach can be used in tandem with numerical optimization techniques to compress circuits, as these can be used as the gate to evolve each partition.

- I suggest authors provide more discussion around the various steps in the algorithm descriptions of both Magnus/Fer-THRIFT. More specifically, one point I feel needed to be addressed, was how in step 4 of both algorithms, the use of a p -th order product formula does not reduce the scaling of the overall error with respect to α from beyond quadratic to quadratic. Although a closer inspection confirms that step 4 of both algorithms does not incur any penalty on the scaling of the error with respect to α , I do feel providing some intuition or a brief discussion on why this is not the case could be helpful to readers.

We thank the Reviewer for this comment. In the new version of the manuscript we have added more details describing the various steps in the Magnus- and Fer-THRIFT algorithms.

As the generator of the time evolution that we are expanding has a norm $\|\alpha H_1(t)\|$, the error has to scale in the same way in time and $\|\alpha\|$; otherwise we could scale time by some arbitrary parameter L and the Hamiltonian by a parameter $1/L$, getting an error that scales with this arbitrary parameter for the exact same evolution.

- In proof of Theorem 4: I believe the left-hand-side of equation (20) was supposed to be $\|\Omega(\alpha, t) - \Omega^{[k]}(\alpha, t)\|$, reflecting the difference between the truncated Magnus series $\Omega^{[k]}(\alpha, t)$, and the complete series $\Omega(\alpha, t)$. I am not sure why $\|R_{[k]}(\Omega(\alpha, t))\|$ appears on the left-hand-side of equation (20) instead. Similarly, I believe the bounds on the integral in equation (21) need to range from 0 to 1 instead of 0 to s . Changing the upper bound from s to 1 makes the left-hand side of equation (22) consistent with the right-hand side.

We thank the Reviewer for this comment and for spotting the typo in Eq. (21). The integral should range from 0 to 1 . We corrected this typo in the new version of the manuscript. We have also explicitly added the connection between the remainder $R_{[k]}$ and the difference $\|\Omega(\alpha, t) - \Omega^{[k]}(\alpha, t)\|$ for clarity.

- I believe the proof of Appendix A.3 is based on the assumption that $\{a_v\}$ are independent of α . Is that a necessary assumption? Is relaxing this assumption meaningful? and does it provide to avoid the nogo?

If the $\{a_v\}$ and therefore the quantity Δ in eq. (88) depend on some parameter β then this can be varied independently of the value of α in the Hamiltonian. With this being the case one could choose the value of β which gives the best upper bound on the value of Δ . However, the result shows that whatever this is, it cannot be zero so the error will always have α^2 scaling.

- I found using $H_1(t)$ to denote H_1 in the interaction picture a bit confusing (below eq 2 and after). Maybe using a tilde?

We thank the referee for pointing out this potential ambiguity. Following their suggestion, in the new version of the manuscript we have replaced $H_1(t)$ with $\tilde{H}_1(t)$ whenever the interaction picture is used.

Reviewer #2 (Remarks to the Author):

We thank the Reviewer for their feedback. Please see the corresponding report for detailed replies.

Reviewer #3 (Remarks to the Author):

I have read the manuscript with interest. Product formulas are central to my work and advancements in their performance and scope are important for quantum computing (and beyond).

My understanding of the main result is that it is a new product formula construction particularly well suited to the simulation of Hamiltonians of the form $H = H_0 + \alpha H_1$, with α small. The authors refer to the method as THRIFT. (The acronym is a bit forced but I'll give it a pass).

While the construction is obtained from an interaction picture representation -- which requires time evolution under time-dependent Hamiltonians -- the result is a specific product formula for time-independent Hamiltonian simulation, namely that of Eq. (8). This product formula includes a product over easy-to-implement exponentials from the decomposition of H_1 , as well as direct exponentials over H_0 . The exponential over H_0 will also need to be decomposed as a product formula, except for the special cases where it is already fast-forwardable (as in the examples they consider). They also demonstrate that the construction works for high-order formulas.

The appeal of this construction is that we obtain an extra factor of α in the error compared to more standard methods like the Suzuki hierarchy. This can be significant enough in practice to motivate the use of this method. The results are then expanded using techniques to decompose the time-ordered exponential. The authors refer to these as Magnus-THRIFT (based on the Magnus operator) and Fer-THRIFT (based on results by Fer to express time-ordered exponential as a product formula). These methods are more complicated, but achieve even better scaling with α for higher orders.

The paper concludes with numerical experiments on simple spin models to evaluate regimes where each of these methods is most appealing. While the numerical results are a bit challenging to parse, my general impression is that the techniques from this work become appealing whenever long time evolutions with extremely low error are required.

Overall I find that this paper introduces a useful result for the field that is well presented and, to the best of my ability, all derivations and theorems are correct. I will consider these results in my work if applications arise that meet the conditions where these methods become attractive.

We would like to thank the Reviewer for their positive comments on our work. Below we will provide detailed responses to the various criticisms and questions.

In passing, we would like to quickly address the Reviewer's comment about the numerical experiments in the last part of our manuscript (we will provide more details in the responses

below). We are grateful to the Reviewer for making us aware of the potential misunderstanding that might arise from the interpretation of our numerical simulations. In particular, we would like to stress that THRIFT formulas provide significant advantages over standard formulas in a wide range of regimes and neither long-time evolution nor low error are essential. Indeed, as we discuss in the response to points 2 and 5 below (and in the new paragraph at the end of Section 4.1), THRIFT formulas outperform standard formulas also in regimes of practical relevance in current near-term quantum computers. In order to highlight this point further and improve the readability of Section 4, we decided to move the simulations of the Fermi-Hubbard model to the Appendix.

Having said that, unfortunately my assessment is that Nature Communications is not the right fit for this work. Instead, I view this work as a "rapid communication": a clever and useful technique to keep in the arsenal for people working on applied quantum computing. In fact, my main criticism of the manuscript is that it contains too much information that dilutes the main results. Pages 1-5 contain the bulk of important results. The Magnus and Fer extensions could be cited as results, with derivations left for an appendix. In particular, I find that the numerical experiments don't add much extra value (and I generally love numerics!). One key numerical study for a representative system that draws clearer conclusions about the regime of applicability would have likely sufficed. I'm not recommending that the authors re-write the manuscript this way, but rather use this example to make the point that core of results and contributions don't evidently meet the standards of Nature Communications, and could otherwise find a better home in a different journal.

We thank the reviewer for bringing to our attention this issue with our manuscript. Following their suggestions we decided to move the discussion of the Fer-THRIFT algorithm and of numerical results for the Fermi-Hubbard model to the Supplemental Material in order to convey our core results more effectively. On the other hand, we believe that the Magnus-THRIFT (as well as Fer-THRIFT) algorithm, despite not being of particular advantage for the systems and regimes we considered in the numerical sections, introduce powerful ideas which may contribute to the development of novel formulas. This is not merely hypothetical as there has been a surge of interest in Magnus-expansion-inspired formulas soon after our work. See for instance [arXiv:2403.13889](https://arxiv.org/abs/2403.13889) and [arXiv:2404.02966](https://arxiv.org/abs/2404.02966). Therefore, we think that keeping it in the main text is more appropriate. We also believe that it is important to illustrate that our results lead to improvements over known techniques for more than one system (and for a system which is not efficiently simulable classically for large sizes), so we have continued to include our results for the Heisenberg model in the main text.

As we highlighted above and explain in more detail below (and in the new version of our manuscript), our formulas represent a significant improvement over standard Trotter formulas in many interesting regimes (including long time evolution and large system sizes but also in regimes of immediate practical relevance for near-term quantum simulations). As our results show, for certain systems, we believe that THRIFT is the current most efficient time-dynamics simulation method by a significant factor.

Furthermore, we believe that the perturbative approach we introduce in our work has the potential to stimulate the development of novel efficient formulas for the simulation of systems beyond the ones we considered. As also pointed out by the first reviewer, developing efficient algorithms for quantum simulation is a crucial area of research to harness the full potential of both near-term and fault-tolerant quantum computers. We are convinced that our results will be of interest to researchers working on both applied and theoretical aspects of quantum computation and simulation, and hence respectfully disagree with the referee about whether this work meets the standards of Nature Communications.

The following are more minor points that I share with authors in case they find them useful:

*1. I'm not sure about the title. It seems to focus more on *how* the main result is derived rather than on describing the main result -- product formulas with lower error for "near-integrable" Hamiltonians.*

We thank the referee for this comment. After careful consideration, we decided to keep the original title of the manuscript. The main reason we believe the current title is the most appropriate one is that, even though our formulas have been derived in the interaction picture assuming a small α (which, as the reviewer pointed out, corresponds to a nearly integrable regime), from a practical point of view we showed that for quantum spin models, these formulas are also advantageous in the intermediate ($\alpha \lesssim 1$) and large α regimes as well (see the regions with $\alpha > 1$ in Figs. 1 and 4 of the manuscript). In light of this, our formulas represent an efficient tool for simulating non-integrable systems (far from their integrable points), such as the 2D transverse-field Ising on a square lattice and the 1D Heisenberg model with random fields. Therefore, we believe it would not be appropriate to highlight either the near-integrability of the models or the perturbative regime in the title of our manuscript. Furthermore, as discussed in a reply to the first referee and also in the next reply below, our method can be used together with numerical compression of the circuit corresponding to a partition of the Hamiltonian. In this case the small parameter becomes the relative norm of the parts in the partition. We would like to thank the reviewer for making us aware that this message was not stressed properly in the previous version of the manuscript. We have added a new sentence in the Introduction to highlight this important point.

2. It would be helpful to draw clearer conclusions on the regimes where this method is most appealing. Even better, it would strengthen the results significantly to showcase that this product formula approach significantly reduces cost of implementation for applications of practical importance. (I'm wondering about this myself. Not sure I can identify where these ideas would have the strongest impact).

We thank the Reviewer for this suggestion that helped us in further clarifying the advantages of our improved product formulas. First of all, we would like to point out that we expect THRIFT algorithms to be of practical relevance in the simulation of a large variety of models with different energy scales. As we highlight in the new version of the manuscript, we believe that microscopic models of quantum magnetism are ideal candidates. While textbook quantum spin models, such

as the Ising or Heisenberg ones, often capture many aspects of the physics of realistic quantum magnets, more complicated models containing several perturbative corrections are often required to obtain a precise description of experimental results. A paradigmatic example is the quasi-1D ferromagnet CoNb_2O_6 , which represents a close realization of the 1D transverse-field Ising model. However, to fully describe the exotic physics occurring near its critical point, one has to consider an extended model with additional weak perturbations to the standard Ising model, which breaks the integrability of the latter and requires numerical investigations [see Coldea et al., *Science* **327**, 5962 (2010), *Phys. Rev. B* **83**, 020407 (2011), and Fava et al., *PNAS* **117**, 25219 (2020)]. Due to the different energy scales present in such microscopic models, we are confident our formulas have the potential to significantly reduce the resources required for investigating the time dynamics of models of relevance in the field of quantum magnetism. Furthermore, we believe that the perturbative approach we develop may work equally well for other classes of systems and will pave the way to new research addressing the dynamics of physical systems involving different energy scales. We would also like to highlight that this approach interfaces nicely with numerical techniques aiming at the compression of quantum circuits. Here, given some classical budget, one aims to generate a description of a circuit in a compressed form using e.g tensor networks. This compressed circuit can be used as the “large” part of the Hamiltonian, from which different extensions can be studied systematically using THRIFT.

In the new version of the manuscript we have also included an example of a simulation in a more practical scenario (i.e., for intermediate time and not too small errors) which can be realized in current quantum computers. We show the THRIFT algorithms we developed in our work would allow us to perform better simulation with limited resources. The figure below (Fig. 3 in the new version of the paper) shows the out-of-equilibrium dynamics of a 1D transverse-field Ising model consisting of 29 sites.

The system is initially prepared in the state $|\downarrow_1 \dots \downarrow_{14} \uparrow_{15} \downarrow_{16} \dots \downarrow_{29}\rangle$, with $|\uparrow\rangle = (|\uparrow\rangle + |\downarrow\rangle)/\sqrt{2}$ (i.e., all the spins but the central one are initialized in the ground state of the field Hamiltonian $H_0 = h \sum_i Z_i$), and evolved according to $H = J \sum_{\{i\}} X_i X_{i+1} + h \sum_i Z_i$. The exact time evolution is shown in the left-hand plot, while the central and the right-hand plots show time dynamics obtained via 2nd-order Trotter and THRIFT formulas,

respectively. The number of steps in both cases is fixed to 15, which is a realistic estimate for the circuits that can be implemented with state-of-the-art quantum computers [see, eg, Kim et al., Nature **618**, 500 (2023)]. Clearly, THRIFT allows us to obtain a better approximation of the dynamics of the system for longer times using the same number of steps and two-qubit gates.

3. It's not so clear where the inequality in Eq. (4) comes from. Perhaps there is a succinct way to help the reader follow the derivation better.

We thank the reviewer for making us aware of this issue. In the revised version of the manuscript we added a few additional comments in Eq. (4) and the following paragraph.

4. In Prop. 2, how are the parameters u_j chosen in practice? This was not very clear.

We thank the reviewer for this comment. In the revised version of the manuscript we now explain how these parameters are chosen in practice.

5. The numerics sections consider very small errors, below $1e-6$ and all the way down to $1e-15$. In my experience this is overkill in practice. Said differently, applications where extremely low errors are needed are not typically where it becomes appealing to use product formulas compared to other algorithms for Hamiltonian simulation (and even less clear that quantum computers are attractive since this implies similarly low error rates). It would strengthen the results to better justify why such low errors are considered and meaningful

We agree with the reviewer that the small errors appearing in Figures 1 and 4 might be a bit misleading. As we wrote in the response to point 2 above (and in the corresponding section in the manuscript), we have now made it clear that the formulas we introduced in our work are capable of outperforming standard formulas in practically any error regime. We agree that errors below $1e-6$ are often of no practical relevance. As we discuss in the new version of the manuscript, the small errors in Figures 1 and 4 are required to allow for a fair comparison between all the formulas we considered. In particular, the total number of Trotter steps is fixed in order to allow one step of the most expensive formula we considered (see Tables 2 and 3 in the main text). For the time interval we considered in Figures 1 and 4, this results in very small approximation errors. On the other hand, Figure 2 and Figure 5 show that, for larger system sizes and evolution times, THRIFT formulas outperform standard formulas even in a regime with a more realistic error around $1e-2$. Moreover, the newly added Figure 3 further clarifies the advantages of using THRIFT in regimes of practical relevance.